# Real World Use of Antidiabetic Drugs in the Years 2011–2017: A Population-Based Study from Southern Italy

**DOI:** 10.3390/ijerph17249514

**Published:** 2020-12-18

**Authors:** Ylenia Ingrasciotta, Maria Paola Bertuccio, Salvatore Crisafulli, Valentina Ientile, Marco Muscianisi, Luca L’Abbate, Maurizio Pastorello, Vincenzo Provenzano, Alessandro Scorsone, Salvatore Scondotto, Gianluca Trifirò

**Affiliations:** 1Department of Biomedical and Dental Sciences and Morphofunctional Imaging, University of Messina, 98125 Messina, Italy; mp.bertuccio@gmail.com (M.P.B.); salcrisafulli@unime.it (S.C.); vientile@unime.it (V.I.); luca.labbate@unime.it (L.L.); 2Department of Clinical and Experimental Medicine, University of Messina, 98125 Messina, Italy; marcom1987@gmail.com; 3Palermo Local Health Unit, Department of Pharmacy, 90127 Palermo, Italy; dipfarmaceuticoasppa@gmail.com; 4Regional Referral Centre for Insulin Pump Implantation and Diabetes, Civic Hospital, Partinico, 90047 Palermo, Italy; vincenzoprovenzano54@gmail.com (V.P.); alexscorsone@gmail.com (A.S.); 5Department of Epidemiologic Observatory, Health Department of Sicily, 90127 Palermo, Italy; salvatore.scondotto@regione.sicilia.it; 6Department of Diagnostic and Public Health, University of Verona, 37134 Verona, Italy; gianluca.trifiro@univr.it

**Keywords:** antidiabetic drugs, real world data, type 2 diabetes mellitus, therapy guidelines, Southern Italy

## Abstract

Diabetes mellitus is a metabolic disease characterized by chronic hyperglycemia. The availability of new antidiabetic drugs (ADs) has led to complex treatment patterns and to changes in the patterns of specific drug utilization. The aim of this population-based study was to describe the pattern of antidiabetic drugs (ADs) use in Southern Italy in the years 2011–2017, in relation to the updated type 2 diabetes mellitus (T2DM) therapy guidelines. A retrospective cohort study was conducted on T2DM patients using data from the Palermo Local Health Unit (LHU) claims database and diabetologist registry. The first-line treatment was investigated and incident treatments were identified and characterized at baseline in terms of demographics, complications, comorbidities, concomitant drugs and clinical parameters. Persistence to AD treatment was also evaluated. During the study period, one-third of first ever ADs users started the treatment with ADs other than metformin, in contrast to guideline recommendations. Among 151,711 incident AD treatments, the male to female ratio was 1.0 and the median age was 66 (57–75) years. More than half (55.0%) of incident treatments discontinued the therapy during the first year of treatment. In Italy, general practitioners (GPs) can only prescribe first-generation ADs, while the prescription of more recently marketed ADs, such as GLP-1RA, DPP4i and SGLT2i, is restricted to diabetologists only, based on a therapeutic plan. The role of GPs in the management of T2DM in Italy should be re-evaluated.

## 1. Introduction

Diabetes mellitus is a metabolic disease characterized by chronic hyperglycemia as a consequence of defects in insulin action, secretion, or both. Diabetes currently affects more than 425 million people over the world, and the number of patients with diabetes is estimated to rise up to 693 million in 2045 [1]. In Europe, 58 million people are affected by diabetes, despite 22 million of these being undiagnosed cases [1]. In Italy, type 2 diabetes (T2DM) accounts for over 90% of diabetes cases, with a prevalence of about 5% [2] in the general population, which further increases in Southern Italian Regions [3]. Specifically, Sicily is among the Italian regions with the largest consumption (around 80 Defined Daily Dose (DDD)/1000 inhabitants/day) of antidiabetic drugs (ADs) [4,5].

The growing number of people with T2DM has a great impact not only in terms of clinical effects, but also in terms of economic burden on the healthcare systems [6]. T2DM is a major risk factor for cardiovascular (CV) diseases, such as stroke, myocardial infarction and peripheral vascular disease, as well as being an independent risk factor for heart failure [7,8]. In patients with diabetes mellitus, CV complications are considered the main cause of morbidity and mortality [9,10]. As mentioned in the 2019 ESC guidelines, the early identification and treatment of comorbidities and factors that increase CV risk can bring great benefits to patients with glucose perturbations [11].

According to the 2012, 2015 and 2018 guidelines of the American Diabetes Association (ADA) and the European Association for the Study of Diabetes (EASD) on the management of hyperglycemia in T2DM, metformin is recommended as a first-line glucose-lowering therapy, if well-tolerated, in addition to lifestyle changes [8,12,13]. When glycemic control is not reached with metformin monotherapy at the maximum tolerated dose, adding a second AD is recommended. In particular, before 2018, the guidelines recommended the prescription of a two-drugs combination, choosing the AD to be added to metformin on the basis of several patient- and disease-specific factors. If glycemic control was not reached within three months of dual therapy, adding another AD (triple therapy) was recommended. Combination with injectable therapy (metformin + insulin + mealtime insulin or glucagon-like peptide-1 receptor agonists (GLP-1RA)) was recommended when glycemic control was not achieved within 3 months of triple therapy [12,13].

On the contrary, the last updated EASD-ADA guidelines recommend either GLP-1RA or sodium–glucose linked transporter 2 inhibitor (SGLT2i) as the second-line treatment in patients affected by established atherosclerotic cardiovascular disease (ASCVD), heart failure or chronic kidney disease (CKD), or in those in which a minimization of weight gain is necessary [8]. In patients without established ASCVD or CKD, if there is a need to minimize hypoglycemia risk, dipeptidyl peptidase-4 inhibitors (DPP4i), GLP-1RA, SGLT2i or glitazones should be added to metformin; finally, only if affordability is a major issue, sulfonylureas or glitazones should be added to metformin. When glycemic control is not achieved with the oral ADs, the previous and the new EASD-ADA guidelines recommend the addition of or the switch to insulin therapy [8,13].

Nevertheless, in Italy, general practitioners (GPs), who play a major role in T2DM management, can only prescribe first-generation ADs, such as metformin, sulfonylureas, glinides, acarbose and glitazones, while the prescription of more recently marketed ADs, such as GLP-1RA, DPP4i and SGLT2i, is restricted to diabetologist only, based on a therapeutic plan [2].

The objective of this population-based study was to explore the real-world use of ADs in a large province of Southern Italy in the years 2011–2017, in relation to recently updated guidelines.

## 2. Materials and Methods

### 2.1. Data Source and Study Design

An observational, retrospective study was conducted from 2011 to 2017 in collaboration with the Sicilian Regional Department for Health Activities and Epidemiological Observatory, the Palermo Local Health Unit (LHU) and the Regional Center of Reference Diabetology and Microinfusion Plant Sicily—Civic Hospital of Partinico (Province of Palermo). Altogether, these centers provided access to fully anonymized data from the claims database and diabetologist registry of Palermo LHU (covering a total population of 1,362,708 people during the study years). As not all T2DM patients were cared for by diabetologists, clinical data from registries were available only for a subsample of the study cohort. The claims database included information on demographics of residents, outpatient pharmacy claims (drug dispensing data), hospital discharges, emergency department (ED) visits and healthcare service payment exemptions.

The diabetologist registry included information on the antidiabetic prescriptions, history of diabetes-related complications and clinical parameters (e.g., glycated hemoglobin, cholesterol, triglycerides, transaminases, albumin extraction rate, creatinine clearance, body mass index (BMI), body weight, waist circumference). The dispensed drugs were coded using the Anatomical Therapeutic Clinical (ATC) classification system and the Marketing Authorization Code (AIC), while comorbidities and complications were coded through the ninth review of the International Classification of Diseases—Clinical Modification (ICD9-CM).

The study protocol was notified to the Ethical Committee of the Academic Hospital of Messina (Prot. N. 0001373 of the 01/24/2018), according to the current national law [14]. The manuscript does not contain clinical studies or patient data. For this type of study formal consent is not required.

### 2.2. Study Population

From the general population of Palermo LHU, all subjects with at least one year of database history and at least one AD dispensed (i.e., prevalent users) during the years 2011–2017 were identified. Of these, all AD users without any dispensing of the same AD in the year prior to the ID (i.e., incident AD users) were then identified. The index date was defined as the date of the first AD dispensing during the study period. The same patient could be included in several AD classes (i.e., incident AD treatment), and multiple index dates (one for each AD that was initiated during follow-up) for the same patient were allowed. All AD users with a diagnosis of T1DM (identified from diabetologist registry and/or from discharge diagnosis database), or with ages less than 20 years and exclusive treatment with insulin, or gestational diabetes or other types of non-T2DM, were excluded.

### 2.3. Study Drugs

All the marketed molecules/classes of ADs included in the study and their ATC codes are listed in Table A1. Specifically, in light of the change of some drug ATCs in 2017, the previous ATCs of these drugs were included as well ((exenatide (A10BX04); liraglutide (A10BX07); lixisenatide (A10BX10); albiglutide (A10BX13); dulaglutide (A10BX14), dapagliflozin (A10BX09); canagliflozin (A10BX11); empagliflozin (A10BX12)) [15].

All available AD dispensing data were identified from the Palermo claims database.

### 2.4. Data Analysis

First, the distribution of the dispensed ADs as first-line treatment was investigated. Only subjects not treated with ADs before the first AD dispensing during the study period (first ever users) were included in this analysis. Second, the prevalence (%) of AD users, stratified by calendar year and AD molecule/class, was also evaluated. Third, among incident treatments, the distribution of previous use of other ADs (i.e., at least one dispensing any time prior to ID) was evaluated and stratified by molecule/class.

Then, a baseline characterization of incident treatments, stratified by antidiabetic drug/class, was performed. The variables examined included patients’ demographics, comorbidities, DM complications, concomitant drugs and clinical parameters (Table A2).

All incident AD treatments with at least 180 days of follow-up and at least one HbA1c measurement within 1 month prior to the ID (i.e., baseline HbA1c value), and another one within the 6th month after ID (follow-up HbA1c), were identified. The difference between the mean follow-up and the baseline HbA1c values, i.e., delta Hb (ΔHbA1c), was calculated and stratified by AD molecule/class.

The same analysis for waist circumference and body weight measurement was performed by comparing values at baseline vs. those measured within 12 months after ID.

Among incident AD treatments, time to discontinuation analysis during the first year of treatment was carried out. For each incident AD treatment, the number of days of continuous therapy from the treatment’s start, based on the DDD and the amount of dispensed ADs, was estimated. Concerning fixed-dose combinations, as DDD does not exist, we assumed that patients received one tablet a day. Persistence to AD treatment was assessed based on the maximum allowed treatment gap, defined as the time between the last day covered by AD treatment (discontinuation date) and the day to the next refill. Patients were considered discontinuers if they had at least one treatment gap exceeding 90 days between the estimated end of exposure of the last drug dispensing and the start of the next one (if any). For discontinuers, the time to discontinuation was calculated as the number of days between ID and the discontinuation date. Antidiabetic drug/class-specific time to discontinuation analysis was performed using a Kaplan–Meier plot. Persistence to insulin was not evaluated since no standard duration of treatment with insulin exists because the actual administered daily doses may vary substantially depending on the health conditions of the patient; therefore, in general, the assessment of adherence and persistence to insulin treatment using claims databases may not be accurate.

Then, discontinuers were re-classified as intermittent users if they received at least one dispensing of the same AD between the discontinuation date and the end of the first year of follow-up. Instead, continuers were re-classified as add-on users if they received at least one additional AD (never prescribed before) between the ID and the end of the first year of follow-up.

All the analyses on drug utilization were carried out only using the claims database covering the whole T2DM population.

Descriptive statistics were used to describe all examined variables. The results are presented as mean ± standard deviation (SD) or median with interquartile range (IQR) depending on the underlying distribution for quantitative variables, and they were summarized by absolute frequencies and percentages for categorical variables.

In order to compare proportions and variations (as %) between the different groups, a Generalized Estimating Equations (GEE) model was fitted, using a binomial and Gaussian link function for binary and continuous variables, respectively. For comparisons within groups, a paired t-test was used. Statistical analyses were performed using SAS 9.2 (SAS Institute, Cary, NC, USA). Generalized Estimating Equations models were performed in R version 3.5.0 using the geepack package (version 1.3-1). The significance level for all statistical tests was set at *p*-value < 0.05.

## 3. Results

During the study period, from a population of about 1.4 million patients registered in Palermo LHU, excluding 2797 patients with T1DM, gestational diabetes or other types of non-T2DM, 151,744 incident T2DM AD treatments were identified. Clinical data from the diabetologist registry were available for a subsample of 48,498 (32.0%) incident AD treatments (Figure 1).

The overall prevalence (%) of AD users was 9.1%, with a stable trend during the study years. However, a slight decrease in metformin + sulfonylureas fixed-dose combination use was observed over time (from 0.6% in 2011 to 0.2% in 2017). On the contrary, a slight increase in the more recently marketed SGLT2i and metformin + SGLT2i fixed-dose combination from 2015 onwards was found. The use of the other first-generation ADs remained almost steady during the study period (Figure A1).

During the observation years, among the 44,692 subjects that were the first ever users of ADs, only two-thirds (N = 29,514; 66.0%) received metformin as the first-line treatment, with the remaining ones mostly receiving insulin (N = 3807; 8.5%) and first-generation ADs, such as sulfonylureas (N = 3638; 8.1%), glinides (N = 2428; 5.4%) and AGIs (N = 2083; 4.7%) (Figure 2).

Non fixed-dose combinations include AD users with more than one AD dispensing at the index date.

Only AD users not treated with any AD before the first dispensing date during the study period were included in this analysis.

Looking at the previous use of ADs, 1581 (37%) and 1769 (61%) patients with GLP1-RA and SGLT2i incident treatments, respectively, had received more than three other ADs before GLP1-RA and SGLT2i were started (Figure 3).

In general, among all incident AD treatments, a male to female ratio of 1.0 was observed, except for incident treatments of metformin + SGLT2i, metformin + glitazones and other oral antidiabetic fixed-dose combinations (male to female ratio = 1.4); the overall median age was 66 (IQR: 57–75) years, except for those undergoing incident treatments of SGLT2i, GLP-1RA, and metformin + SGLT2i fixed-dose combinations, who were significantly younger (i.e., 56 years) (*p*-value < 0.05) (Table 1). Only a very low proportion of incident treatments of SGLT2i and GLP1-RA had a history of heart failure or nephropathy, compared to incident treatments of glinides and insulin (heart failure: 5.1% and 5.9% vs. 10.2% and 11.6%, respectively (*p*-values < 0.001 for both comparisons); nephropathy: 7.7% and 7.4% vs. 11.2% and 12.4%, respectively (*p*-values < 0.001 for both comparisons)).

History of heart failure was observed in 4.3% of incident treatments of glitazones. Hypertension and lipid metabolism disorders were the most common comorbidities (76.6% and 50.7%, respectively), and proton pump inhibitors (55.5%) were the most frequently used concomitant drugs.

Among incident treatments of ADs with available clinical parameters from the diabetologist registry, the baseline HbA1c values were always over the target range, with the highest values being reported for incident treatments of SGLT2i (N = 1221 (79.6%); median (IQR): 8.9% (7.9–9.9%)), insulin (N = 3845 (47.6%); median (IQR): 8.8% (7.7–10.2%)) and a fixed-dose combination of metformin + sulfonylureas (N = 463 (48.2%); median (IQR): 8.7% (7.6–10.0%)) (*p*-value < 0.05) (Table 2). Within 6 months after ID, the variation (%) in HbA1c values was more marked among incident treatments of fixed-dose combination of metformin + SGLT2i (−15.7%), SGLT2i (−12.8%) and insulin (−12.7%), as compared to other AD classes (*p*-values < 0.001) (Figure 4).

All incident AD treatments within at least 6 months of follow-up, with at least 1 HbA1c measurement at baseline and within 6 months from baseline, were included.

The highest baseline body weight as well as waist circumference values were observed among incident treatments of newly marketed ADs, such as GLP1-RA (N = 1127 (59.0%); median (IQR) body weight: 95.0 kg (84.0–107.5 kg); N = 569 (29.8%), median (IQR) waist circumference: 116.0 cm (108.0–123.0 cm)) and fixed-dose combinations of metformin + SGLT2i (N = 438 (79.5%); median (IQR) body weight: 89.0 kg (79.0–102.0 kg); N = 295 (53.5%), median (IQR) waist circumference: 115.0 cm (106.0–120.0 cm)) and SGLT2i (N = 1231 (80.3%); median (IQR) body weight: 87.7 kg (77.5–100.4 kg); N = 669 (43.6%), median (IQR) waist circumference: 112.0 cm (104.0–120.0 cm)) (*p*-values < 0.05) as compared with other AD groups (Table 2). Statistically significant changes in body weight and waist circumference values within one year after ID were observed (Figure A2 and Figure A3). Specifically, the reduction in body weight values was more marked among incident treatments of fixed-dose combination of metformin+SGLT2i (−2.4%), SGLT2i (−1.7%) and GLP1-RA (−1.6%) (*p*-values < 0.001) as compared with other AD groups. Additionally, for waist circumference, the reduction in mean values was more marked among incident treatments of fixed-dose combinations of metformin+SGLT2i (2.1%) and GLP1-RA (−1.6%;) (*p*-values < 0.001).

During the first year of AD treatment, more than half (55.0%) of the 106,484 incident AD treatments included in the persistence analysis discontinued the treatment, with the highest proportions of discontinuers being observed among incident treatments of the metformin + sulfonylureas fixed-dose combination (65.9%) and AGIs (62.2%) (*p*-value < 0.05). Moreover, 52.3% of sulfonylureas users discontinued during the first year of treatment. One-fourth of incident treatments received only one dispensing of AD (i.e., occasional users) during the first year of treatment. On the contrary, the proportion of discontinuers was lower among incident treatments of more recently marketed ADs, such as GLP1-RA and fixed-dose combinations of metformin+SGLT2i and metformin + DPP4i (*p*-values < 0.05 for comparison with other AD groups) (Figure 5).

Overall, within the first year of treatment, 13.9% of incident AD treatments were intermittent users, as the same AD treatment was restarted after discontinuation. The highest percentages of intermittent users were found among metformin users (17.7%) (*p*-value < 0.05). Overall, 20,389 (19.1%) incident AD treatments received an add-on therapy during the first year of treatment, mostly among incident treatments of metformin (29.5%) (Figure A4).

## 4. Discussion

This population-based study investigated the prescribing pattern of ADs in a large province of Southern Italy, which has one of the highest T2DM prevalence rates in Italy. Our results showed that only two-thirds of T2DM patients received metformin as the first-line treatment, in line with guidelines. Around 10% of T2DM patients received as the first-line treatment insulin, which may be appropriate only in cases of marked glycometabolic decompensation or CKD [16]. In particular, in 3–5 CKD stage patients, insulin therapy is appropriate with proper dosage reduction [17].

Instead, more than 20% of T2DM patients were treated with other non-metformin first-generation ADs, such as sulfonylureas, glinides and AGIs, which is in contrast with the updated guidelines unless the use of metformin was not tolerated or contraindicated (e.g., severe renal impairment, that is, CKD stage ≥4). An exploratory analysis showed that among the 336 patients starting the treatment with ADs different from metformin and for which information of the CKD stage was available, 137 (40.8%) subjects had a diagnosis of severe renal impairment (e.g., CKD stage IV–V or dialysis) and were therefore not eligible for the treatment with metformin. However, 11% of incident treatments of glinides had a history of nephropathy; despite the fact that in this clinical condition updated guidelines suggest using SGLT2i or GLP-1RA, glinides are considered safe in advanced renal disease with cautious dosing [8].

Concerning glitazones, our results showed that a relatively low proportion of glitazones users had a history of heart failure despite the fact that they are significantly and consistently associated with a high risk of heart failure [18], and therefore they are not recommended in patients with established New York Heart Association (NYHA) III/IV heart failure [19].

Due to hypoglycemia risk, current treatment guidelines recommend sulfonylureas only as a third-line treatment, and this is mostly driven by affordability issues. However, a population-based study conducted in Ireland in 2015 [20,21] showed that sulfonylureas were used as first-line treatment in 22% of T2DM patients. In another population-based study conducted in Catalonia in 2016 [22], the authors noted an increase in the use of metformin and a decrease in the use of sulfonylureas from 2007 to 2013, and similar results were found in other population-based studies [23,24,25,26]. Finally, the wide use of sulfonylureas and other non-metformin first-generation ADs as observed in our study may be due to the fact that most of the T2DM patients are cared for by GPs only, who are allowed to prescribe only those low-cost ADs.

Moreover, our results showed that users of GLP-1RA or SGLT2i, which are recommended as second-line treatments according to updated guidelines, received four other ADs belonging to different pharmacological classes, as an add-on or switch, before starting treatment with these drugs. Furthermore, these pictures have to be carefully interpreted as updated guidelines recommending the use of SGLT2i as a second-line treatment were published in 2018, while those drugs were marketed only in April 2015 in Italy, thus limiting the assessment of patterns of SGLT2i use in our study. Moreover, our results show that the highest baseline median HbA1c values were reported for incident SGLT2i treatments. Accordingly, the SGLT2i class was the only class among ADs recently marketed that could be prescribed by specialists in Italy without any restriction based on HbA1c values, while GLP1-RA could be prescribed only to patients with HbA1c values between 7.5% and 8.5% [8]. Moreover, within the first six months of therapy, the reduction in HbA1c values was more marked among incident treatments of insulin, SGLT2i and fixed-dose combinations of metformin + SGLT2i than other ADs. This was in line with a meta-analysis conducted by Storgaard et al., showing that SGLT2is, at the recommended daily target doses, were associated with a larger reduction in HbA1c levels in patients with T2DM as compared with other existing oral ADs, especially sulfonylureas and DPP4i [27]. This is especially true when SGLT2is are used in combination with metformin, due to the complementary effects of these drugs, which have different mechanisms of actions [28]. It is known that insulin is the most effective AD for reducing HbA1c levels, and its main advantage is that the HbA1c level’s reduction is dose-dependent [29].

As expected, the highest baseline body weight values, as well as waist circumference values, were observed among incident treatments of more recently marketed ADs, such as GLP1-RA, SGLT2i and fixed-dose combinations of metformin + SGLT2i. This is in line with guidelines, which recommend the prescription of these ADs in patients prioritizing weight loss or weight maintenance [8].

In general, discontinuation may result from intolerance or side-effects, and our results showed that during the first year of treatment, more than half of the patients with incident treatments of ADs, especially AGIs and fixed-dose combinations of metformin + sulfonylureas, discontinued the therapy. This finding is in line with other national/international population-based studies, showing that up to 50% of T2DM patients discontinued AD treatment during the first year of therapy [30,31]. AD treatment discontinuation in the elderly population could be explained by the need to avoid hypoglycemic events that can be fatal in fragile patients; that is why higher glycated hemoglobin levels are allowed in elderly patients [32]. Concerning AGIs, discontinuation was probably due to their known gastrointestinal effects [8,33]. The high proportion of discontinuers among sulfonylureas users as well as users of fixed-dose combinations of metformin + sulfonylureas could be explained by the high risk of symptomatic hypoglycemic events [34]. AD treatment discontinuation is associated with a worsening of glucose metabolism, and therefore with increased glycated hemoglobin values that could determine a series of consequences related to macroangiopathy, microangiopathy and diabetic neuropathy [35]. Such consequences may lead to an increased cardiovascular risk and clinical complications that can negatively impact the NHS.

In our study, 25% of the incident treatments were occasional (only one AD dispensing), especially among AGIs and metformin groups (data not shown). However, the proportion of metformin occasional users could probably be overestimated. In Italy, metformin is very cheap and, according to the “IMS-Health Italy” database, which provides aggregated data on drugs dispensed by private pharmacies and distinguishes between dispensing reimbursed and not reimbursed (e.g., paid directly by patients) by the National Health System, around 20% of metformin is purchased outside the National health Service (NHS), hence not being fully traceable using claims databases. In addition, almost one-third of incident treatments of metformin received an add-on therapy during the first year of treatment, probably because glycemic control was not achieved with metformin monotherapy.

On the contrary, lower proportions of discontinuers were observed among incident treatments of more recently marketed ADs, such as GLP1-RA, DPP4 and metformin+DPP4i, which are generally well tolerated, and are associated with weight loss and low risk of hypoglycemia [36]. Moreover, patients treated with these ADs are cared for by diabetologist (as GPs cannot prescribe those drugs), who can be more effective in motivating patients to continue AD treatment, especially as periodic visits are planned to assess therapy and eventually renew therapeutic plans.

### Strengths and Limitations

The main strengths of this study are the large size of the study cohort and the availability of the claims plus diabetologist registry from the same catchment area for a period of 8 years.

On the other hand, some study limitations warrant caution. First, clinical data from the diabetologist registry were limited to a low sample of the study cohort with likely more advanced or complicated diabetes mellitus. In Italy, only the diabetologist may prescribe more recently marketed ADs, such as GLP-1RA, DPP4i and SGLT2i, while GPs can prescribe only first-generation ADs [37].

Second, the persistence to treatment was assessed on the basis of DDD for non-fixed dose combinations, but clearly situations in which some patients have to intensify the therapy could occur, and therefore DDD could not reflect the doses used in the real world. This is the case of insulin: as no standard duration exists because it depends on the health condition of the patient, insulin was excluded from the persistence analysis (Figure 5) and from the analysis of the distribution of continuers, add-on, discontinuers and intermittent users (Figure A4). Moreover, the doses of sulfonylureas, usually prescribed by GPs, are substantially variable (from half to three tablets a day, as specified in the summary of product characteristics). For this reason, we cannot rule out that persistence to sulfonylureas may be partly underestimated. However, the proportion of discontinuers among users of sulfonylureas was not significantly higher than discontinuers of other Ads, and it is well known that the hypoglycemic effects of sulfonylureas may trigger discontinuation. Concerning fixed-dose combinations, as a DDD does not exist, we assumed that the dosing regimen was one tablet a day. However, in the case of fixed-dose combination of metformin + sulfonylureas, the dose, again, is related to patient tolerance or the patient’s renal function. For this reason, discontinuation could be overestimated.

Third, not all AD dispensing can be completely traced, as 16% of ADs dispensed in Palermo LHU were reported as being purchased outside the NHS during the study years, with the highest percentages for lowest-costing Ads, such as metformin (cost = EUR 3.00; estimated purchase outside the NHS = 20%). However, in Italy, it is unlikely that a patient after receiving a diagnosis of diabetes mellitus and a first AD prescription by the GP/diabetologist will purchase a drug outside the NHS in private pharmacy, without any reimbursement, and consequently the first AD dispensing is missing. As such, it is highly unlikely that the first AD drug dispensing goes untraced. However, this could at least partly explain the very high proportion of intermittent users/discontinuers that could directly purchase a low-cost AD in a private pharmacy without receiving the prescription from the GP/specialist.

Finally, our findings cannot be fully generalized to the whole Italian general population, as the study was restricted to a LHU of Southern Italy.

## 5. Conclusions

In a general population from Southern Italy, a large use of non-metformin first-generation ADs, as both first- and second-line treatments and with high short-term discontinuation rates, was observed. This is likely due to the fact that about 70% of T2DM patients in the Palermo LHU were cared for only by GPs, who may prescribe only first-generation ADs, despite updated guidelines recommending as second-line therapies more recent and higher-cost drugs, such as DPP4i, GLP1-RA and SGLT2i. A linkage of claims databases with clinical registries can help rapidly explore the effectiveness and safety issues in real-world settings. In the near future, further analyses re-evaluating the pattern of recently marketed antidiabetic drugs in relation to the updated type 2 diabetes mellitus therapy guidelines, as well as the role of GPs in the prescription of more recently marketed ADs in Italy, need to be performed. It is also crucial to establish an alliance between physicians and patients aimed at implementing treatment adherence by increasing patient’s knowledge of the possible complications that chronic diseases such as diabetes can cause. New strategies, such as lifestyle modifications, reductions in the major risk factors (e.g., obesity, physical activity, tobacco smoking, alcohol consumption and mental stress) responsible for T2DM, and other educational programs for both physicians and patients, are needed to prevent and/or delay the onset of diabetes and to promote AD therapy adherence in order to avoid, or reduce as much as possible, treatment-related adverse events. This would result in significant reductions in social burden and economic costs.

## Figures and Tables

**Figure 1 ijerph-17-09514-f001:**
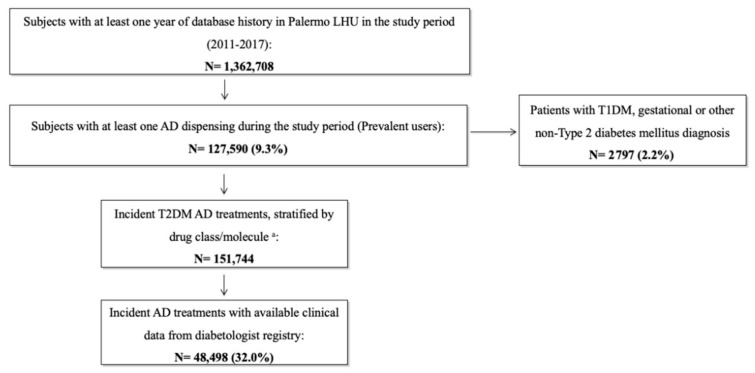
Flow-chart of T2DM patients receiving antidiabetic drugs in Palermo Local Health Unit in the years 2011–2017. Legend: LHU: Local Health Unit; AD: Antidiabetic drug; T1DM: Type 1 diabetes mellitus; T2DM: Type 2 diabetes mellitus. ^a^ Subjects without any AD dispensing of the same drug within one year prior to the treatment start date, i.e., index date (ID). The same patient could be included in different AD classes; for this reason, incident T2DM AD treatments were more numerous than prevalent users.

**Figure 2 ijerph-17-09514-f002:**
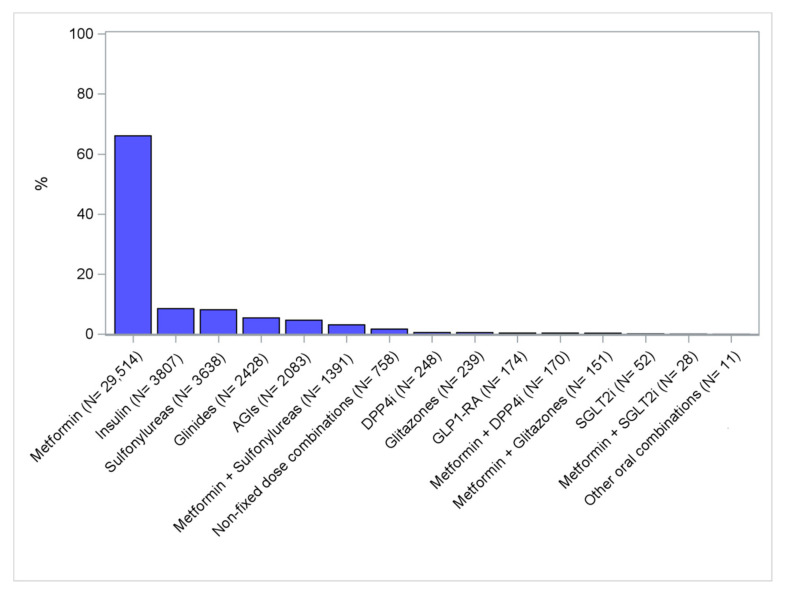
Distribution (%) of the first antidiabetic drug use, stratified by antidiabetic molecule/class. Legend: AGI = alpha-glucosidase inhibitors; DPP4i = dipeptidyl peptidase-4 inhibitors; GLP-1RA = glucagon-like peptide-1 receptor antagonists; SGLT2i = sodium–glucose linked transporter 2 inhibitors; Other oral combinations = fixed-dose combinations of glitazones + DPP4i and glitazones + sulfonylureas.

**Figure 3 ijerph-17-09514-f003:**
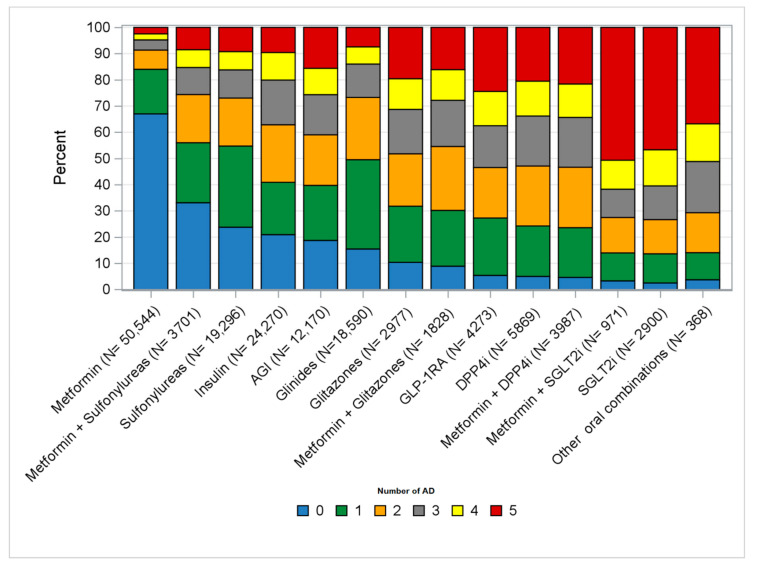
Distribution (%) of the number of ADs dispensed any time prior to the treatment start among incident treatments. Legend: AGI = alpha-glucosidase inhibitors; DPP4i = dipeptidyl peptidase-4 inhibitors; GLP-1RA = glucagon-like peptide-1 receptor antagonists; SGLT2i = sodium–glucose linked transporter 2 inhibitors; Other oral combinations = fixed-dose combinations of glitazones + DPP4i and glitazones + sulfonylureas. Only number of ATCs belonging to a different antidiabetic molecule/class as compared to the molecule/class that was dispensed at the index date.

**Figure 4 ijerph-17-09514-f004:**
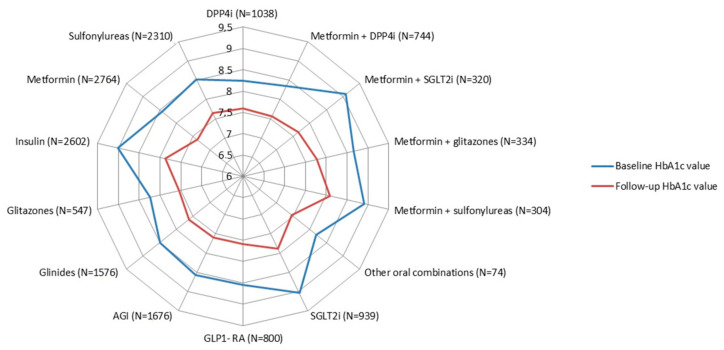
Comparison between the mean HbA1c values (%) at baseline and within six months after ID, stratified by AD molecule/class. Legend: AGI = alpha-glucosidase inhibitors; DPP4i = dipeptidyl peptidase-4 inhibitors; GLP-1RA = glucagon-like peptide-1 receptor antagonists; SGLT2i = sodium–glucose linked transporter 2 inhibitors; Other oral combinations = fixed-dose combinations of glitazones + DPP4i and glitazones + sulfonylureas; ID = index date. In subjects ≤ 70 years old, glycated hemoglobin target value < 7%; in subjects > 70 years old, glycated hemoglobin target value ≤ 8%.

**Figure 5 ijerph-17-09514-f005:**
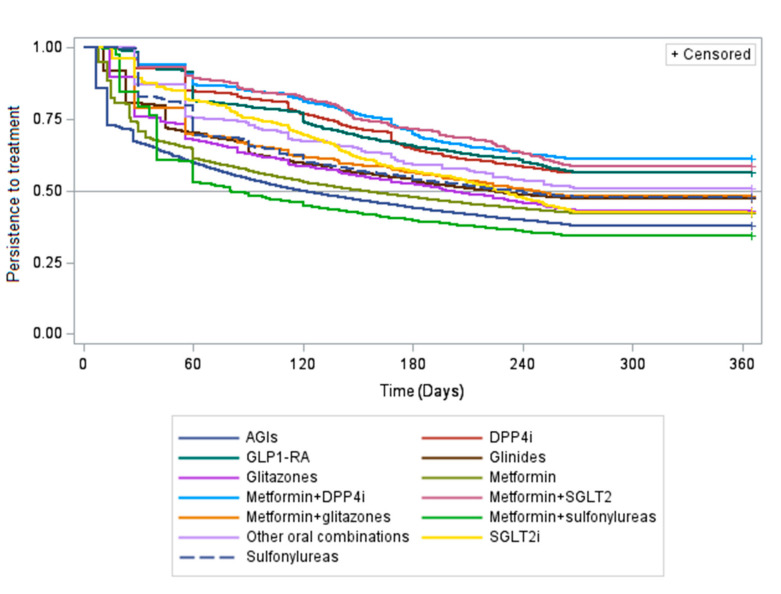
Time to discontinuation of AD treatment during the first year among incident treatments, stratified by antidiabetic molecule/class. Legend: AGI = alpha-glucosidase inhibitors; DPP4i = dipeptidyl peptidase-4 inhibitors; GLP-1RA = glucagon-like peptide-1 receptor antagonists; SGLT2i = sodium–glucose linked transporter 2 inhibitors; Other oral combinations = fixed-dose combinations of glitazones + DPP4i and glitazones + sulfonylureas.

**Table 1 ijerph-17-09514-t001:** Baseline characteristics of incident AD treatments in Palermo LHU during the years 2011–2017.

	Metformin N = 50,544 (%)	Insulin N = 24,270 (%)	SU N = 19,296 (%)	Glinides N = 18,590 (%)	AGI N = 12,170 (%)	DPP4i N = 5869 (%)	GLP-1 RA N = 4273 (%)	Glitazones N = 2977 (%)	SGLT2i N = 2900 (%)	Metformin + DPP4i N = 3987 (%)	Metformin + SGLT2i N = 971 (%)	Metformin + Glitazones N = 1828 (%)	Metformin + SU N = 3701 (%)	Other Oral Combinations N = 368 (%)	Total N = 151,744 (%)
**Males**	24,852 (49.2)	12,643 (52.1)	9716 (50.4)	9314 (50.1)	5835 (47.9)	3079 (52.5)	2308 (54.0)	1589 (53.4)	1547 (53.3)	2224 (55.8)	569 (58.6)	1067 (58.4)	1848 (49.9)	213 (57.9)	76,804 (50.6)
**Median Age** **(Q1–Q3)**	64 (55–72)	69 (58–77)	68 (60–76)	70 (61–77)	66 (58–74)	65 (58–73)	57 (50–64)	64 (56–71)	57 (50–63)	62 (55–69)	56 (49–61)	63 (55–70)	71 (63–79)	63 (56–70)	66 (57–75)
Age ranges—N (%)															
**<45**	6820 (13.5)	2192 (9.0)	1045 (5.4)	731 (3.9)	715 (5.9)	260 (4.4)	594 (13.9)	218 (7.3)	310 (10.7)	233 (5.8)	140 (14.4)	126 (6.9)	195 (5.3)	21 (5.7)	13,600 (9.0)
**45–64**	21,864 (43.3)	7866 (32.4)	8015 (41.5)	5994 (32.3)	4814 (39.6)	2437 (41.5)	2790 (65.3)	1491 (50.2)	2061 (71.1)	2231 (56.0)	686 (70.6)	1006 (55.0)	1389 (37.5)	186 (50.5)	62,830 (41.4)
**65–84**	20,076 (39.7)	12,372 (51.0)	9396 (48.7)	10,650 (57.3)	6170 (50.7)	3056 (52.1)	880 (20.6)	1216 (40.8)	527 (18.1)	1503 (37.7)	145 (14.9)	665 (36.4)	1896 (51.2)	157 (42.7)	68,709 (45.3)
**≥85**	1784 (3.5)	1840 (7.6)	840 (4.4)	1215 (6.5)	471 (3.9)	116 (2.0)	9 (0.2)	52 (1.7)	2 (0.1)	20 (0.5)	–	31 (1.7)	221 (6.0)	4 (1.1)	6605 (4.3)
Comorbidities ^a^ —N (%)															
**Heart failure**	2129 (4.2)	2813 (11.6)	1159 (6.0)	1888 (10.2)	834 (6.9)	501 (8.5)	254 (5.9)	129 (4.3)	149 (5.1)	156 (3.9)	26 (2.7)	75 (4.1)	216 (5.8)	14 (3.8)	10,343 (6.8)
**Atrial fibrillation**	2001 (4.0)	2164 (8.9)	1024 (5.3)	1519 (8.2)	667 (5.5)	339 (5.8)	184 (4.3)	99 (3.3)	110 (3.8)	129 (3.2)	18 (1.9)	53 (2.9)	194 (5.2)	7 (1.9)	8508 (5.6)
**Hypertension** ^b^	34,762 (68.8)	19,040 (78.5)	15,412 (79.9)	15,711 (84.5)	9940 (81.7)	4918 (83.8)	3469 (81.2)	2264 (76.0)	2304 (79.4)	3181 (79.8)	754 (77.7)	1373 (75.1)	2858 (77.2)	280 (76.1)	116,266 (76.6)
**Lipid metabolism disorders** ^b^	20,356 (40.3)	11,960 (49.3)	10,361 (53.7)	10,303 (55.4)	7165 (58.9)	3964 (67.5)	2706 (63.3)	1792 (60.2)	1950 (67.2)	2,715 (68.1)	616(63.4)	1,102 (60.3)	1754 (47.4)	243 (66.0)	76,987 (50.7)
**COPD**	2008 (4.0)	2014 (8.3)	928 (4.8)	1331 (7.2)	596 (4.9)	281 (4.8)	196 (4.6)	101 (3.4)	94 (3.2)	126 (3.2)	23 (2.4)	46 (2.5)	209 (5.6)	8 (2.2)	7961 (5.2)
**Dementia** ^b^	583 (1.2)	659 (2.7)	274 (1.4)	366 (2.0)	197 (1.6)	95 (1.6)	17 (0.4)	35 (1.2)	7 (0.2)	32 (0.8)	5 (0.5)	4 (0.2)	67 (1.8)	5 (1.4)	2346 (1.5)
**Liver disease**	1769 (3.5)	1598 (6.6)	737 (3.8)	811 (4.4)	530 (4.4)	294 (5.0)	241 (5.6)	121 (4.1)	167 (5.8)	177 (4.4)	47 (4.8)	59 (3.2)	159 (4.3)	13 (3.5)	6723 (4.4)
**Anemia**	1737 (3.4)	2300 (9.5)	945 (4.9)	1380 (7.4)	671 (5.5)	364 (6.2)	177 (4.1)	139 (4.7)	109 (3.8)	154 (3.9)	32 (3.3)	64 (3.5)	216 (5.8)	18 (4.9)	8306 (5.5)
**Hyperuricemia/Gout** ^b^	7965 (15.8)	5752 (23.7)	4128 (21.4)	5161 (27.8)	2970 (24.4)	1452 (24.7)	870 (20.4)	591 (19.9)	516 (17.8)	633 (15.9)	143 (14.7)	307 (16.8)	727 (19.6)	74 (20.1)	31,289 (20.6)
Complications ^e^—N (%)															
**Nephropathy**	1737 (3.4)	3007 (12.4)	1168 (6.1)	2087 (11.2)	1016 (8.3)	688 (11.7)	315 (7.4)	236 (7.9)	223 (7.7)	195 (4.9)	41 (4.2)	89 (4.9)	217 (5.9)	22 (6.0)	11,041 (7.3)
**Retinopathy**	1252 (2.5)	1308 (5.4)	993 (5.1)	906 (4.9)	907 (7.5)	486 (8.3)	355 (8.3)	194 (6.5)	395 (13.6)	337 (8.5)	106 (10.9)	112 (6.1)	159 (4.3)	28 (7.6)	7538 (5.0)
**Neuropathy**	770 (1.5)	491 (2.0)	362 (1.9)	380 (2.0)	251 (2.1)	157 (2.7)	126 (2.9)	55 (1.8)	89 (3.1)	105 (2.6)	26 (2.7)	31 (1.7)	62 (1.7)	11 (3.0)	2916 (1.9)
**Cardiovascular diseases** ^c^	5676 (11.2)	5206 (21.5)	2993 (15.5)	3561 (19.2)	2036 (16.7)	1283 (21.9)	737 (17.2)	384 (12.9)	689 (23.8)	710 (17.8)	161 (16.6)	212 (11.6)	546 (14.8)	41 (11.1)	24,235 (16.0)
**Cerebrovascular diseases** ^d^	5059 (10.0)	4796 (19.8)	2550 (13.2)	3204 (17.2)	1638 (13.5)	967 (16.5)	474 (11.1)	313 (10.5)	421 (14.5)	525 (13.2)	113 (11.6)	154 (8.4)	522 (14.1)	42 (11.4)	20,778 (13.7)
**Diabetic foot**	188 (0.4)	249 (1.0)	128 (0.7)	140 (0.8)	151 (1.2)	70 (1.2)	57 (1.3)	29 (1.0)	50 (1.7)	43 (1.1)	19 (2.0)	21 (1.1)	42 (1.1)	4 (1.1)	1191 (0.8)
Concomitant drugs ^f^ —N (%)															
**Proton pump inhibitors**	23,168 (45.8)	15,376 (63.4)	11,071 (57.4)	11,850 (63.7)	7335 (60.3)	3770 (64.2)	2351 (55.0)	1638 (55.0)	1601 (55.2)	2,232 (56.0)	477 (49.1)	924 (50.5)	2156 (58.3)	208 (56.5)	84,157 (55.5)
**Anticoagulants** ^g^	4572 (9.1)	4828 (19.9)	2287 (11.9)	2991 (16.1)	1541 (12.7)	728 (12.4)	445 (10.4)	293 (9.8)	249 (8.6)	378 (9.5)	70 (7.2)	165 (9.0)	506 (13.7)	31 (8.4)	9084 (12.6)
**Antiplatelet agents** ^h^	14,577 (28.8)	10,797 (44.5)	7956 (41.2)	8801 (47.3)	5626 (46.2)	2838 (48.4)	1577 (36.9)	1158 (38.9)	1183 (40.8)	1,676 (42.0)	347 (35.7)	681 (37.3)	1537 (41.5)	154 (41.9)	58,908 (38.8)
**Systemic glucocorticoids**	6556 (13.0)	4914 (20.2)	2905 (15.1)	3132 (16.8)	1688 (13.9)	667 (11.4)	405 (9.5)	354 (11.9)	262 (9.0)	374 (9.4)	72 (7.4)	195 (10.7)	613 (16.6)	40 (10.9)	22,177 (14.6)
**Antidepressants**	4593 (9.1)	2557 (10.5)	2061 (10.7)	2112 (11.4)	1383 (11.4)	617 (10.5)	436 (10.2)	326 (11.0)	260 (9.0)	375 (9.4)	95 (9.8)	170 (9.3)	442 (11.9)	46 (12.5)	15,473 (10.2)
**Drugs for neuropathic pain ^i^**	2128 (4.2)	1648 (6.8)	1286 (6.7)	1326 (7.1)	963 (7.9)	497 (8.5)	388 (9.1)	219 (7.4)	236 (8.1)	315 (7.9)	87 (9.0)	124 (6.8)	263 (7.1)	33 (9.0)	9513 (6.3)
**Anti-osteoporotic drugs**	1815 (3.6)	1085 (4.5)	1010 (5.2)	1026 (5.5)	747 (6.1)	340 (5.8)	112 (2.6)	150 (5.0)	82 (2.8)	186 (4.7)	29 (3.0)	72 (3.9)	181 (4.9)	12 (3.3)	6847 (4.5)

Legend: AGI = alpha-glucosidase inhibitors; DPP4i = dipeptidyl peptidase-4 inhibitors; GLP-1RA = glucagon-like peptide-1 receptor antagonists; SGLT2i = sodium–glucose linked transporter 2 inhibitors; SU = sulfonylureas; COPD = chronic obstructive pulmonary disease; DM = diabetes mellitus; ID = index date; IQR = interquartile range. The other oral combinations include fixed-dose combinations of glitazones + DPP4i and glitazones + sulfonylureas. ^a^ Evaluated any time prior to ID and identified looking at ICD9-CM codes from discharge diagnosis database or codes from exemptions from healthcare service co-payment database. ^b^ Dementia, hypertension, lipid metabolism disorders and gout were identified by looking at ICD9-CM codes from the discharge diagnosis database, or codes from exemptions from the healthcare service co-payment database or the specific ATC codes from the outpatient drug dispensing database. ^c^ This category includes: angina pectoris, ischemic heart disease, acute myocardial infarction. ^d^ This category includes: transient cerebral ischemia and stroke. ^e^ Evaluated any time prior to ID and identified by looking at ICD-9 codes from discharge diagnosis database or diabetologist registry or codes from exemptions from the healthcare service co-payment database. ^f^ Evaluated within three months prior to ID and identified by looking at ATC codes from outpatient drug dispensing. ^g^ This category includes: vitamin K antagonists, heparins, thrombin direct inhibitors, factor Xa direct inhibitors, fibrinolytic agents. ^h^ This category includes: platelet aggregation inhibitors. ^i^ This category includes: Pregabalin, Gabapentin and Duloxetin.

**Table 2 ijerph-17-09514-t002:** Baseline clinical parameters of incident AD treatments in Palermo LHU during the years 2011–2017.

	Metformin N = 12,230	Insulin N = 8081	SU N = 6587	Glinides N = 5532	AGI N = 4759	DPP4i N = 2411	GLP-1 RA N = 1909	Glitazones N = 1299	SGLT2i N = 1533	Metformin + DPP4i N = 1713	Metformin + SGLT2i N = 551	Metformin + Glitazones N = 778	Metformin + SU N = 961	Other Oral Combinations N = 154	Total N = 48,498
Clinical parameter: median [IQR] (Proportion (%) of incident AD treatments with ≥ 1 measurement within 3 months prior to ID)															
**Glycated hemoglobin (target range: 7–8%) ***	8.0 [7.0–9.2] (38.0)	8.8 [7.7–10.2] (47.6)	8.2 [7.4–9.3] (52.8)	8.2 [7.2–9.3] (44.7)	8.2 [7.4–9.4] (52.7)	8.0 [7.5–8.7] (59.3)	8.3 [7.7–9.2] (58.3)	8.0 [7.1–9.1] (60.7)	8.9 [7.9–9.9] (79.6)	8.2 [7.6–8.8] (59.7)	8.3 [8.0–9.9] (79.3)	8.5 [7.6–9.7] (63.5)	8.7 [7.6–10.0] (48.2)	8.0 [7.6–8.9] (68.8)	8.3 [7.4–9.4] (49.5)
**Cholesterol (target range: 120–220 mg/dl)**	184.0 [155.0–216.0] (34.1)	172.0 [145.0–202.0] (41.0)	173.0 [147.0–204.0] (46.8)	173.0 [146.0–202.0] (40.0)	170.0 [144.0–200.0] (46.8)	171.0 [144.0–199.0] (51.5)	170.0 [142.0–199.0] (51.8)	175.0 [149.0–203.0] (53.3)	168.0 [142.0–197.0] (68.0)	164.0 [141.0–193.0] (52.4)	171.0 [147.0–195.0] (68.2)	176.0 [149.0–202.0] (59.4)	176.0 [148.5–210.5] (42.9)	177.5 [159.0–213.0] (63.6)	174.0 [147.0–204.0] (43.7)
**SGOT (target range: 1–35 UI/l)**	20.0 [15.0–27.0] (22.6)	18.0 [14.0–25.0] (28.1)	19.0 [15.0–26.0] (33.5)	18.0 [14.0–25.0] (25.5)	19.0 [15.0–25.0] (33.3)	19.0 [15.0–25.0] (38.0)	21.0 [16.0–28.0] (37.8)	19.0 [15.0–25.0] (39.1)	20.0 [15.0–28.0] (49.4)	18.0 [15.0–25.0] (38.2)	20.0 [15.0–28.0] (48.8)	19.0 [15.0–28.0] (32.0)	18.0 [15.0–25.0] (24.3)	18.0 [14.0–22.5] (44.2)	19.0 [15.0–26.0] (30.1)
**SGPT (target range: 1–43 UI/l)**	23.0 [16.0–36.0] (23.9)	21.0 [14.0–32.0] (29.8)	22.0 [16.0–33.0] (35.2)	21.0 [15.0–30.0] (27.8)	21.0 [15.0–31.0] (36.0)	21.0 [15.0–31.0] (39.6)	26.0 [18.0–39.0] (39.4)	21.0 [15.0–31.0] (41.3)	24.0 [17.0–36.0] (50.0)	22.0 [16.0–32.0] (39.4)	25.0 [17.0–36.0] (51.0)	25.0 [17.0–37.0] 39.5)	22.0 [15.0–30.0] (30.7)	20.0 [15.0–28.0] (53.2)	22.0 [16.0–34.0] (32.1)
**Albumin extraction rate (target range: 1, 5–20 µg/min)**	10.9 [5.0–27.0] (11.8)	13.60 [6.0–40.30] (16.5)	12.0 [5.0–33.0] (19.0)	14.0 [5.6–49.0] (14.8)	13.0 [5.3–37.0] (18.2)	13.0 [5.0–37.5] (21.3)	14.0 [6.0–47.5] (22.7)	11.0 [5.0–26.0] (25.1)	12.3 [5.20–41.0] (32.2)	12.8 [5.3–31.0] (23.0)	12.0 [5.5–30.0] (32.3)	12.0 [5.0–34.0] (25.2)	11.8 [5.0–35.0] (17.7)	12.0 [7.0–57.8] (29.9)	12.15 [5.0–36.0] (17.5)
**Creatinine clearance (target range: 70–120 mL/min)**	87.0 [66.0–115.0] (17.2)	78.0 [53.0–108.0] (25.7)	84.0 [60.0–111.0] (27.0)	67.0 [48.0–96.0] (25.1)	79.0 [56.0–107.0] (27.1)	72.0 [51.0–99.0] (33.5)	113.0 [89.0–145.0] (30.4)	78.0 [57.0–108.0] (32.7)	102.5 [81.0–133.5] (45.9)	88.0 [68.0–107.0] (30.9)	111.0 [89.0–139.0] (52.1)	95.0 [73.0–119.0] (32.5)	80.0 [58.0–110.0] (21.3)	81.0 [62.0–108.0] (27.9)	84.0 [60.0–114.0] (25.7)
**BMI (target range: 18.5–24.9 Kg/m** ^**2**^ **)**	29.9 [26.7–34.0] (37.5)	29.3 [25.8–33.1] (45.7)	30.0 [26.6–33.7] (51.1)	29.4 [26.2–33.3] (42.8)	30.1 [26.8–34.0] (48.7)	28.9 [25.8–32.4] (57.3)	34.5 [31.0–39.1] (57.5)	28.4 [25.2–32.5] (58.1)	32.4 [28.9–36.7] (79.3)	29.4 [26.6–32.9] (57.5)	32.5 [29.0–36.7] (78.9)	30.5 [27.5–34.0] (59.6)	30.1 [26.5–33.6] (43.4)	28.8 [24.6–32.5] (61.0)	30.1 [26.6–34.0] (47.8)
**Waist circumference (target values: women: <** ** 80cm;men: <94cm)**	108.0 [100.0–116.0](14.8)	106.0 [98.0–115.0] (17.2)	107.0 [99.0–116.0] (19.9)	107.0 [100.0–116.0] (14.6)	109.0 [100.0–118.0] (16.6)	103.0 [97.0–112.0] (24.8)	116.0 [108.0–123.0] (29.8)	102.0 [94.0–112.0] (23.2)	112.0 [104.0–120.0] (43.6)	105.0 [99.0–113.0] (29.0)	115.0 [106.0–120.0] (53.5)	107.0 [100.0–115.0] (26.6)	104.0 [98.0–114.0] (15.1)	100.0 [92.0–109.0] (25.3)	108.0 [100.0–117.0] (19.4)
**Body weight (kg)**	80.0 [70.0–93.0] (38.7)	78.0 [67.5–90.0] (47.3)	79.0 [69.0–91.0] (52.6)	77.0 [68.0–89.0] (44.3)	80.0 [70.0–91.7] (50.3)	77.0 [66.8–87.0] (58.4)	95.0 [84.0–107.5] (59.0)	75.7 [66.0–88.0] (59.7)	87.7 [77.5–100.4] (80.3)	80.0 [71.0–90.0] (58.7)	89.0 [79.0–102.0] (79.5)	82.6 [72.0–93.0] (60.9)	79.0 [69.0–91.0] (44.8)	77.2 [66.5–86.3] (62.3)	80.0 [70.0–92.5] (49.2)

Legend: AGI = alpha-glucosidase inhibitors; DPP4i = dipeptidyl peptidase-4 inhibitors; GLP-1RA = glucagon-like peptide-1 receptor antagonists; SGLT2i = sodium–glucose linked transporter 2 inhibitors; SU = sulfonylureas; DM = diabetes mellitus; IQR = interquartile range; SGOT = serum glutamic oxaloacetic transaminase; SGPT = serum glutamate-pyruvate transaminase; BMI = body mass index. The other oral combinations include fixed-dose combinations of glitazones + DPP4i and glitazones + sulfonylureas. * In subjects ≤ 70 years old, glycated hemoglobin target value < 7%; in subjects 70 years old, glycated hemoglobin target value ≤ 8%.

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
