# Peer review of "Real World Use of Antidiabetic Drugs in the Years 2011–2017: A Population-Based Study from Southern Italy"

_ijerph, 2020, doi:10.3390/ijerph17249514_

Round 1
Reviewer 1 Report
The study is interesting, due to the large sample that was analyzed and the focus given.
Manuscript can improve in several ways. Some points are described.
1) Establish whether the drugs used for the treatment of diabetes are prescribed to patients according to a basic scheme of drugs that the health department acquires, or is related to formal studies that show the improvement of patients when consuming certain medicines.
2) Consider the inclusion of information related to natural treatments, such as the use of medicinal plants, if the population uses any of these treatments and if this could modify the results obtained.
3) Emphasize if there are actions that are helping to reduce Diabetes mellitus in the population studied
4) Describe how this type of study impacts the health area to improve the comprehensive process to control the disease
5) The cost of the disease is very high, so it is recommended to strengthen this type of foundation to propose strategies for its treatment.
6) Emphasize the results of the graphs shown according to the analysis of the variables.
Author Response
Response to Reviewer 1 Comments
The study is interesting, due to the large sample that was analyzed and the focus given.
Manuscript can improve in several ways. Some points are described.
1) Establish whether the drugs used for the treatment of diabetes are prescribed to patients according to a basic scheme of drugs that the health department acquires, or is related to formal studies that show the improvement of patients when consuming certain medicines.
Response 1: We thank the reviewer for this comment. If the reviewer refers to patients starting antidiabetic treatment with drugs other than metformin, as we mentioned in the discussion section, this is probably due to a marked glycometabolic decompensation or contraindication to the treatment with metformin (e.g. chronic kidney disease). Moreover, as we stated in the “Strengths and limitations” paragraph, not all metformin dispensing can be completely traced, so exposure to metformin could be underestimated. In general, antidiabetic drugs should be prescribed according to the guidelines; it not depends to a basic scheme of drugs that the health department acquires.
2) Consider the inclusion of information related to natural treatments, such as the use of medicinal plants, if the population uses any of these treatments and if this could modify the results obtained.
Response 2: We thank the reviewer for bringing up this point. However, use of medicinal plants is not recommended by guidelines and we are not able to trace them using Italian claims databases.
3) Emphasize if there are actions that are helping to reduce Diabetes mellitus in the population studied
Response 3: We thank the reviewer for this comment. As also suggested by the second reviewer, we have updated the Conclusions section, adding that “New strategies, such as (e.g. lifestyle modifications, reduction of the major risk factors (e.g. obesity, physical activity, tobacco smoking, alcohol consumption and mental stress) responsible for T2DM and other educational programs for both physicians and patients,) are needed to prevent and/or delay the onset of diabetes and to promote AD therapy adherence in order to avoid, or reduce as much as possible, treatment-related adverse events. This would result in significant reduction in social burden and economic costs” (Page 18; lines: 453-458).
4) Describe how this type of study impacts the health area to improve the comprehensive process to control the disease
Response 4: As reported in the Strengths and limitations section, this population-based study investigated the prescribing pattern of ADs in a large province of Southern Italy, covering more than 1.3 million of inhabitants for a period of 8 years. Moreover, in the Conclusions section we have added that a linkage of claims databases with clinical registries can rapidly explore effectiveness and safety issues in real world settings (Page 18; lines: 446-448).
5) The cost of the disease is very high, so it is recommended to strengthen this type of foundation to propose strategies for its treatment.
Response 5: We thank the reviewer for bringing up this point. Unfortunately, the economic burden of the disease was not the focus of the paper; however, as mentioned in the comment above, we have updated the Conclusions section indicating the potential strategies useful to prevent and/or delay the onset of diabetes, the onset of adverse events and consequently the economic impact on the NHS.
6) Emphasize the results of the graphs shown according to the analysis of the variables.
Response 6: We thank the reviewer for this comment. We have discussed the results represented in the graphs focusing above all on the statistically significant between-groups differences.

Reviewer 2 Report
This is a study that compares the prescription of ADs in an area of southern Italy against the recently updated guidelines.
The Data analysis section is cumbersome and long. You should try to summarize it so that the reader continues reading the article. It can be provided at length in supplementary material.
Within the results section, it was to be expected that the most used drug would be metformin if the GPs could not prescribe new hypoglycemic drugs.
One of the most interesting results is the finding of high short-term discontinuation rate. These dropouts from treatment can have long-term consequences.
In Table 2 there does not appear to be any difference between the characteristics of the patients and the hypoglycemic treatment used.
It is a very complex study with a very poor conclusion: "The role of GPs in the prescription of more recently marketed ADs in Italy needs to be re-evaluated". Solutions to the problems encountered were expected to be sought: 1) high short-term discontinuation rate; 2) low use of new drugs; 3) lack of follow-up of the guidelines.
Author Response
Response to Reviewer 2 Comments
This is a study that compares the prescription of ADs in an area of southern Italy against the recently updated guidelines.
1) The Data analysis section is cumbersome and long. You should try to summarize it so that the reader continues reading the article. It can be provided at length in supplementary material.
Response 1: We thank the reviewer for this suggestion. We have summarized the Data Analysis section and listed and described all the covariates in the new Table S2.
2) Within the results section, it was to be expected that the most used drug would be metformin if the GPs could not prescribe new hypoglycemic drugs.
One of the most interesting results is the finding of high short-term discontinuation rate. These dropouts from treatment can have long-term consequences.
Response 2: We thank the reviewer for bringing up this points. Concerning the first point, as reported in the discussion and limitation sections, the proportion of metformin users could be underestimated because of the proportion of patients that purchased metformin outside the NHS (e.g.: 20%) in Palermo LHU. Moreover, AD patients could start the treatment with insulin in case of marked glycometabolic decompensation or CKD. Moreover, 16% of ADs dispensed in Palermo LHU has been reported as being purchased outside the NHS during the study years; most of this proportion probably concerns first generation ADs because more recently marketed ADs, such as GLP-1RA, DPP4i and SGLT2i is restricted to diabetologist only, based on a therapeutic plan and for this reason it is always traced.
Concerning the discontinuation, we have added the following statements in the Discussion section: “AD treatment discontinuation in the elderly population could be explained by the need to avoid hypoglycemic events that can be fatal in fragile patients; that is why higher glycated hemoglobin levels are allowed in elderly patients [31] […] AD treatment discontinuation is associated with a worsening of glucose metabolism and therefore to increased glycated hemoglobin values that could determine a series of consequences related to macroangiopathy, microangiopathy and diabetic neuropathy [34]. Such consequences may lead to an increased cardiovascular risk and clinical complications that can negatively impact the NHS” (Page 17; lines: 379-389).
3) In Table 2 there does not appear to be any difference between the characteristics of the patients and the hypoglycemic treatment used.
Response 3: We agree with the reviewer. For this reason, in the Results section we described only statistically significant differences, e.g. glycated hemoglobin values.
4) It is a very complex study with a very poor conclusion: "The role of GPs in the prescription of more recently marketed ADs in Italy needs to be re-evaluated". Solutions to the problems encountered were expected to be sought: 1) high short-term discontinuation rate; 2) low use of new drugs; 3) lack of follow-up of the guidelines.
Response 4: We thank the reviewer for this comment. As also suggested by the first reviewer, we have updated the Conclusions section, adding that “In the next future, further analyses evaluating the pattern of recently marketed antidiabetic drugs in relation to the updated type 2 diabetes mellitus therapy guidelines as well as the role of GPs in the prescription of more recently marketed ADs in Italy need to be re-evaluated. It is also crucial to establish an alliance between physicians and patients aimed at implementing treatment adherence by increasing patient's knowledge of the possible complications that chronic diseases such as diabetes can cause. New strategies, such as lifestyle modifications, reduction of the major risk factors (e.g. obesity, physical activity, tobacco smoking, alcohol consumption and mental stress) responsible for T2DM and other educational programs for both physicians and patients, are needed to prevent and/or delay the onset of diabetes and to promote AD therapy adherence in order to avoid, or reduce as much as possible, treatment-related adverse events. This would result in significant reduction in social burden and economic costs” (Page 18; lines: 448-458).
